# Complementing Solutions to Optimization Problems via Crowdsourcing on Video Game Plays

**Mariano Vargas-Santiago** [1,*,†], **Raúl Monroy** [1,†], **José Emmanuel Ramirez-Marquez** [2,†],
**Chi Zhang** [3,†], **Diana A. Leon-Velasco** [4,†] **and Huaxing Zhu** [5,†]

1    Tecnologico de Monterrey, School of Engineering and Science, Carretera al Lago de Guadalupe Km. 3.5,
     Atizapán, Estado de Mexico 52926, Mexico; raulm@tec.mx
2    Enterprise Science and Engineering Division, Stevens Institute of Technology,
     School of Systems & Enterprises, Hoboken, NJ 07030, USA; jmarquez@stevens.edu
3    School of Economics and Engineering, Beijing University of Technology, Beijing 100124, China;
     czhang@bjut.edu.cn
4    Department of Mathematics, Universidad Autónoma Metropolitana, Cuajimalpa 05348, Mexico;
     dleon@correo.cua.uam.mx
5    Department of Industrial Engineering, Tsinghua University, Beijing 100084, China;
     zhuhx15@mails.tsinghua.edu.cn
*    Correspondence: mariano.v.santiago@tec.mx
†    These authors contributed equally to this work.

**Abstract:** Leveraging human insight and intuition has been identified as having the potential for the improvement of traditional algorithmic methods. For example, in a video game, a user may not only be entertained but may also be challenged to beat the score of another player; additionally, the user can learn complicated concepts, such as multi-objective optimization, with two or more conflicting objectives. Traditional methods, including Tabu search and genetic algorithms, require substantial computational time and resources to find solutions to multi-objective optimization problems (MOPs). In this paper, we report on the use of video games as a way to gather novel solutions to optimization problems. We hypothesize that humans may find solutions that complement those found mechanically either because the computer algorithm did not find a solution or because the solution provided by the crowdsourcing of video games approach is better. We model two different video games (one for the facility location problem and one for scheduling problems), we demonstrate that the solution space obtained by a computer algorithm can be extended or improved by crowdsourcing novel solutions found by humans playing a video game.

**Keywords:** combinatorial optimization; facilities planning and design; multiple objective programming; genetic algorithms; scheduling

## 1. Introduction

Leveraging human insight and intuition has been identified as having potential for the improvement of traditional algorithmic methods [1,2], especially in the context of computationally hard problems. Numerous optimization problems, including combinatorial problems (with nonlinear objective functions, nonlinear constraints, or simulation-based objective functions) and multiple-objective problems, are common in practical situations but also are computationally hard [3–5]. In many cases, solving these problems may quickly become intractable.

Although there has been extensive work on the development of heuristics to obtain quasi-optimal solutions for many such problems, to the best of our knowledge, there have been no studies that address possible human-machine-based approaches. Thus, this work aims to demonstrate that human

participants can provide valuable insight to the solution of hard problems and shows the viability of a hybrid human/computer approach against a purely algorithmic approach.

In the literature, we have found that humans have been asked to play video games for crowdsourcing solutions to computationally hard problems [6–12]. Using crowdsourcing in video games to solve difficult problems is an excellent tool for increasing user participation, combining these techniques it is the brainpower of the people that becomes a powerful "processor". This video game approach has been used in a vast range of applications, such as marketing, health, biology, biochemistry, education and even linguistics [7–12].

Although diverse heuristic-based algorithms, such as Tabu search, or a multi-objective evolutionary algorithm (MOEA), can be used in an attempt to solve similar problems, we have identified an opportunity niche for cases in which such algorithms produce void answers, get stuck in a local optimal value and therefore exhibit gaps in their solution space, or have a heuristic efficiency that is inferior to human responses. We demonstrate through the crowdsourcing video game approach that players can solve large-scale incomplete problems, learn complex concepts, such as multi-objective optimization, and complement mechanically found solutions.

In this paper we propose to model two different optimization problems through video games. One is a scheduling problem, and the other is a facility location problem (FLP). We illustrate how players can help to solve these problems. For example, in a single-objective optimization scheduling problem, the goal is to minimize the maximum completion time (makespan). Here a greedy best-first (GBF) search algorithm starts with an initial solution generated by a given heuristic. Algorithms generally do not sacrifice an immediate gain for a more significant long-term gain; in contrast, human users may sacrifice an immediate gain to consequently achieve an overall better schedule. Therefore, users can complement and improve the algorithm solution quality by reducing the overall makespan.

A similar result was observed for FLP, a MOP that arises in many practical situations. Users achieve reasonable trade-offs among objectives, unexpectedly finding solutions overlooked by the machine. We have found that these solutions can be used to extend or improve a genetic algorithm (GA) in terms of solution quality.

This video game approach is also beneficial in other ways; for instance, players are challenged to beat the score of other players and are willing to beat the highest reported score (consequently improving known solutions). Hence, human participants are easily motivated to find optimal solutions. So, our goal is to complement solutions to optimization problems via crowdsourcing video game plays.

Although we have gathered a significant number of candidate solutions, building an algorithm or heuristic with strategies of users is not a simple task for at least two reasons. First, insight and intuition of humans help them understand, as they play, how to solve problems that computers cannot yet solve. Second, translating this insight and intuition is challenging, as users seldom describe the steps taken for their solution or reasoning process. One approach for dealing with these issues is applying machine learning techniques; however, such research is beyond the scope of this work and hence is left for further work.

This paper makes the following contributions:

1. A study demonstrating while playing video games humans perspicacity helps them solve complex problems. While video games have been widely used to gather solutions to complex problems, only a handful of them have been used for helping in the solution of optimization problems.
2. The use of crowdsourcing to a video-game so as to collect solutions to complex problems. These solutions are up to complement the coverage of popular techniques, such as Tabu search or genetic algorithms.
3. A collection of Video games through which the general public can learn about a variety of optimization problems. While playing our video game, people also learn about optimization, both single objective and multi-objective.

The remainder of this paper is organized as follows. In Section 2, we present related work in the theory of human-based computing and provide a brief background regarding games with a purpose. Section 3 details the implementation of the game, the background of select NP problems, and the methodology of play-testing experiments. Section 4 presents the obtained results and a corresponding discussion. Conclusions and future work are given in Section 5.

## 2. State of the Art

In this section, we give an overview of human-based computation, video games with a purpose, and crowdsourcing features as the present work represents a combination of these techniques.

### 2.1. Human-Based Computation

In human-based computation, computers delegate to humans calculation tasks that computers cannot yet solve [13]. For instance, interactive evolutionary computation is used for diverse applications such as graphic arts, speech processing, music generation, knowledge acquisition, robotics, and geophysics [1]. However, traditional algorithms, such as heuristic-based and genetic algorithms, perform poorly in these applications.

In human-based computation, humans and computers can take two roles, selector or innovator. Depending on the assigned role for the human (H) or computer (C), computation methods are divided into four categories:

- *HC*: The human is the innovator, and the computer is the selector. For instance, in computer-assisted design, human participants submit a design that is then evaluated by the computer, in terms, for example, of performance.
- *CH*: The computer is the innovator, and the human is the selector. For instance, an interactive genetic algorithm may produce a population of individuals, which is afterwards presented to a human, who is responsible for evaluating the fitness of the individuals.
- *CS*: The computer takes both roles; this approach corresponds to traditional genetic algorithms.
- *HH*: The human takes both roles. For example, in human-based genetic algorithms, humans provide candidate solutions and then evaluate them.

In particular, in the experiments for scheduling and FLP (see Sections 3.2 and 3.3), we use a computer program to measure the total cost for crowd-sourced candidate solutions. These methods are classified into six dimensions [14]: motivation, quality, aggregation, human skill, participation time and cognitive load. They are defined as follows:

- *Motivation* answers the question: why do contributors choose to help?
- *Quality*: answers the question: how does the system cope with fraud?
- *Aggregation*: the question: How does the system combine local computations made by individual participants to solve the global problem?
- *Human skill*: answers the question: what is the special skill that humans possess that makes them valuable in the system?
- *Participation time*: answers the question: what is the minimum amount of time a user must invest to participate?
- *Cognitive load*: answers the question, how much mental energy does the task take?

Later on in the text, we will demonstrate that we use video games to model optimization problems to help users to feel *motivated* to solve such problems. *Quality* is enforced by having controlled playtests. For *human skills*, the games aim at leveraging human spatial intuition. Levels are kept short to decrease the *participation time* per dataset, encouraging the player to spend more time overall. In this way, we aim to avoid burdening the user with a substantial *cognitive load*.

Human-based computation has a vast range of applications in diverse areas. For instance, human-computer optimization has been explored through interactive graphical interfaces, specifically

to address the capacitated vehicle routing with time windows problem [2]. Studies have also evaluated the optimal methods for strength aggregation of human and computer synergy [15]. In many cases, this technique has been used for military purposes; for example, cooperating teams consisting of humans and computer agents are more effective than a team of only humans in military command and control tasks [16]. In other application domains, we find a combination of both strengths (genetic algorithms and human intelligence), including information retrieval systems [17] and object-oriented software design and development [18].

The above-mentioned research has drawbacks compared with video games. First, users need training for expertly guiding an algorithm, which reduces the potential use of the population at large. Second, a substantial cognitive load is required; thus, users must anticipate the next step of the computer.

*2.2. Games for Solving Problems*

Several video games known as games with a purpose [19], have been developed to take advantage of gamer prowess for solving computationally hard problems, which are often modeled as puzzles. These games address issues related to a vast range of applications, such as biology, biochemistry, and linguistics.

BioGames [7] is an image-matching game that trains players to recognize malaria-infected red blood cells, resulting in an engaging training program for medical student personnel. This video game serves as a method for crowdsourcing labeling data for training machine learning algorithms. BioGames demonstrates the ability of a crowd of untrained participants to achieve a diagnosis of diseases with an accuracy comparable to that of an expert.

Phylo [8] is a color-matching game that encourages players to optimize the alignment of nucleotide sequences, minimizing the number of mutations required to produce different species from an ancestor, producing phylogenic trees, and thus elucidating the process of evolution between related species whose DNA has been sequenced. Phylo focuses on an NP-complete problem called multiple sequence alignment [20], which demonstrates how a difficult problem can be adapted to an entertaining game.

Foldit is a game in which users discover ways to fold proteins, while EteRNA http://www.eternagame.org/web/about/ is a game in which users discover ways to fold RNA, allowing non-expert players to predict how a highly complex molecule will fold. In EteRNA, the players vote on each other's proposals, and their predictions are later synthesized and tested for accuracyy. Foldit has been successful in allowing players to express their strategies through a recipe, letting them automate actions they commonly make in sequence analysis. After observing these recipes, new strategies are applied to enhance the performance of previously developed algorithms [21]. Foldit also provides a method for encoding and transmitting the insight gained by users as they play the game by supplying the users with a scripting language. crowdsourcing, therefore, is a means for discovering new algorithms and heuristics for solving complex problems.

EyeWire [10,11] is an application developed for mapping the outlines of neurons in three dimensions, starting from 2D image slices obtained from the retina. EyeWire asks players to fill in the trace of a neuron in 3D space by following the edges of the 2D images. The game is supplemental to an algorithmic approach, often prompting the player to focus their efforts in areas where the algorithm cannot derive the correct answer. From the player input, maps of complete neurons and their connections have been obtained from such images, enhancing our understanding of the function of eye vision. EyeWire is a successful example of humans and computers cooperating to solve a single problem, focusing on the strengths of one component in regions of the problem space where the other component would be ineffective.

Google Image Labeler (GIL), based on a previous game called ESP [12], asks players to label images obtained from the web. Two players, unable to communicate with one another, must propose the same label for an image. When both players agree on a label, called a matching label, the label is used as metadata to enhance the image of the company search service. TagATune [22,23] is

similar to GIL in its goals and gameplay, obtaining metadata for audio files, such as the genre, instrument, and vocalist gender. These games incorporate a multiplayer component to increase their attractiveness and entertainment value, which doubles as a validation mechanism. Because the players are encouraged to agree and the only way to communicate is through the mechanics of the game, wrong solutions—whether submitted intentionally or otherwise—can be easily detected and discarded.

### 2.3. Crowdsourcing

Despite the numerous works around for solving NP-hard problems, some questions remain open. For instance, There are instances of NP-hard problems that a computer is not up to solve in reasonable time. One wonders: why not use a human-machine approach? Why should one continue using automatic methods only? We as researches can use emerging trends such as crowdsourcing and video games.

Even though it has not a universal definition, in many works of the academia and of industry, crowdsourcing has the following elements—the *crowd*, the *crowdsourcer*, the *crowdsourced task* and the *crowdsourcing platform* [24–26]. The crowd usually constitutes a large number of participants; however, largeness occurs when the crowd participating in a crowdsourcing activity is enough to fulfill a task [24]. We have found cases where two participants are referred to as the crowd [27], yet, they comply with the previous definition. The crowdsourcer can be a company or institution or even a group of researchers. The crowdsourced task is usually a task that cannot be solved easily by computers or it can be a task that is not cost effective for a company [24]. The crowdsourcing platform must have a set of desirable characteristics, such as: *online environment*, *user-driven*, *diversity*, and *open call* (for more information on each of these elements, please refer to Reference [24]). For our research, we have built a crowdsourcing platform trying to adhere as much as possible to most of the aforementioned characteristics.

One of the most known crowdsourcing platforms is Amazon's Mechanical Turk, where companies and individuals post a difficult problem (often related to business and marketing but it can be of any kind), and ask the crowd to solve it, giving a monetary incentive in return. As shown in a mobile crowdsourcing application, many users are unwilling to participate if doing so requires a high energy or considerable bandwidth cost [28]. However, entertainment is often a sufficient reward; some people are naturally drawn to games of the puzzle genre because they feel intellectually challenged and are satisfied upon a successful puzzle solution. Puzzle games also encourage young people to interact with computers in a more productive manner [29].

Crowds can also leverage both formulation and data points in machine science applications [30]. When using data generated by untrained users, however, the quality of their contributions is a significant concern. Attempts to automatically determine which contributions are suitable include ranking human participants according to their consistency and giving more weight to data submitted by the highest-ranked participants [31] or establishing a voting system such that a group of scientists can assign scores to the data. This approach generates profiles on the skills of each scientist according to a set of information quality dimensions such as accuracy, completeness, reputation, and relevance [32].

The present work is, to our knowledge, the first to apply crowdsourcing and gamification to solve several variants of scheduling problems as well as the FLP. The only similar work we have found is Reference [33], where authors modeled FLP and crowdsourced it as a video game, called *SolveIt*, and used it to compare their findings against other approach, called *Cooper*. There, the goal of the authors was to prove users can provide better results than those reported by Cooper. Our approach differs from theirs because we offer a complementing approach to a broader range of methods rather than providing a way of comparison. We provide a supplementary method that combines mechanically found solutions with those humans have found via a gamification technique.

## 3. Implementation

In this section, we give a general description of the two video games we have implemented.

### 3.1. Programming Language and Deployment

We have built two video games, one for solving three variants of scheduling, a single-objective optimization problem, and one for solving facility location, a multi-objective optimization problem. They have been implemented in the Python programming language using Kivy http://kivy.org. Test subjects participated playing on a Dell Latitude 5490 with Intel Core i7-8650U CPU, running Windows 10 as OS.

For the present study, we recruited 20 game players, graduate and undergraduate students. The undergraduate students were not part of a particular group, pursuing different degrees, including, among others, computer science, mathematics, biology, social sciences, and law. The age of undergraduate students ranged from 19 to 23. We also recruited graduate students from two different programs, one in computer science, and the other in applied mathematics. These students were in the age range 30–35.

### 3.2. Game 1: Scheduling Problems

#### 3.2.1. Background for the Scheduling Problems

Scheduling problems consist of finding assignments, called schedules, for a set of jobs on a set of machines. Each machine can process at most one job at a time, and each job can be processed on at most one machine at a time. There is a wide range of scheduling problems, with different characteristics and levels of complexity (http://www2.informatik.uni-osnabrueck.de/knust/class/). They can be divided into a three-field classification, in terms of the nature of the machines, that of the jobs, and optimality criteria [34]. The first field has two components: the machine type, and the number of machines, m. The machine type, in turn, can be either:

1. Identical parallel machines, where any machine can process any job without impacting processing time.
2. Open shop, where each job has an operation for each machine, but where the order of operations within a job is irrelevant.
3. Job shop, where any machine can process any operation, but where operations must be completed in an specified order.

The second field characterizes the execution of jobs; it can be either:

1. Preemption, which indicates whether the processing of a job can be interrupted and later resumed.
2. Presence of limited resources, which means that the processing of a job is constrained to the use of limited resources.
3. Precedence constraints, which imposes ordering constraints in the execution of jobs.
4. Bound on the number of operations, which indicates there is a constant upper bound on the number of operations per job.
5. Bound on processing time, which conveys whether the processing time for each job is constant, bounded, or unbounded.

The third field, optimality criteria, provides a way to compare various schedules. Optimality criteria can be based on either the maximum value (optimizing the performance of each machine) or the sum (optimizing the performance of all machines overall) of metrics such as completion time, lateness, or tardiness.

For the scheduling game, participants solved a variety of example problems with different numbers of machines and with jobs that consist of more than one operation. Operations could be executed only on a specific machine.

#### 3.2.2. Game Description

In all our three scheduling problems, a complete schedule is shown via a single common visualization and interface, leaving only small differences in the moves available to the players

for each variant. In the implementation of the game, we were inspired by the 1982 puzzle video game Tetris, designed by Alexey Pajitnov. In Tetris, a player is given control of a collection of falling tetrominoes; that is shapes constructed from four square tiles. There, the goal of a player is to timely pack the tetrominoes on a rectangular board, while minimizing the height of the resulting pile of tiles.

By contrast, in our game, a player controls one job at a time, aiming at minimizing the length of the schedule, so-called the *makespan*. The video game puts up three consecutive stages, one for each of three scheduling problems, namely: identical parallel machines, open shop scheduling, and job shop scheduling. They are as follows:

**Identical Parallel Machines (IPM):** Players must schedule a job consisting of a single task and need only assign the task to a machine. The task can be moved from one machine to other. Figure 1 convey the user interface of all our three scheduling games. In particular, Figure 1a,b give the player interface for IPM for m = 5 and m = 7, respectively. Despite its simplicity, this version of IPM is not solvable in polynomial time, as can be verified in Reference [35].

**Job Shop Scheduling (JSS):** Players must deal with a job consisting of multiple tasks, one for each machine. Players can choose the machine (a column) to which a task is assigned; however, tasks have to observe a chronological ordering constraint for completion. The game controls allow players to switch a task between machines; hence, users can select the most appropriate machine where to execute a task. Figure 1c,d show the user interface for JSS for m = 5 and m = 7.

**Open Shop Scheduling (OSS):** Players must schedule a job consisting of multiple tasks, one for each machine. Players may choose the chronological order for task execution; however, they have no control as to which machine should process a task. The game interface enables the move of a task to the end of a task queue, enabling players to arrange task execution as they wish. Figure 1e,f show the user interface for a player to solve OSS for m = 5 and m = 7.

On each playthrough, every participant has to play these three games, consecutively. Each scheduling variant is preceded with a tutorial, giving the user a chance to adapt to the game controls.

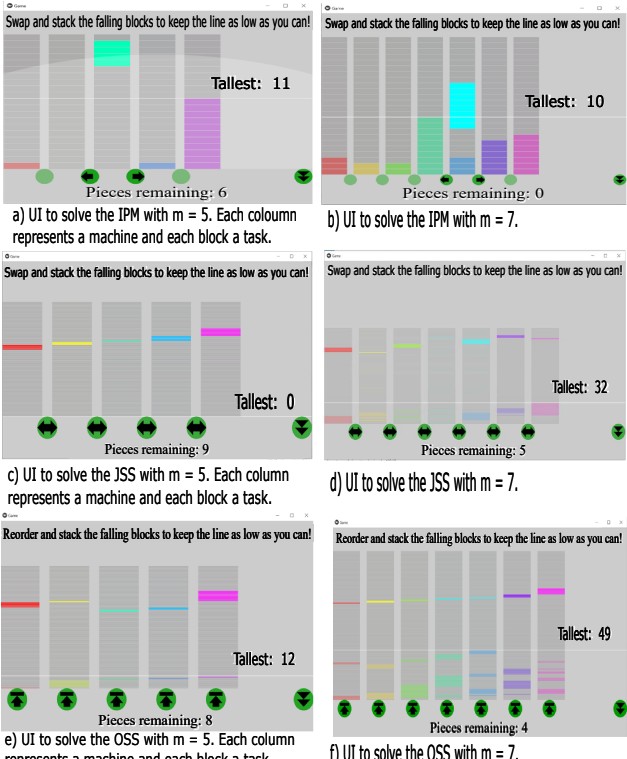

**Figure 1.** User interface for the scheduling game. Here, (**a**,**b**) show the Identical Parallel Machines variant; (**c**,**d**) the Job Shop Scheduling problem variant; and, finally, (**e**,**f**) show the Open Shop Scheduling problem variant.

User Interface Design, Gameplay and Playtesting

The User Interface (UI) design resembles a timetable, or a Gantt chart rotated by 90 degrees, with the time axis running vertically. Each column is assigned a different color to ensure that the players can distinguish them. Falling blocks, that is, tasks that can be controlled by the player at a given time point, have different colors depending on the column in which they are highlighted. Additionally, tasks move downward at a steady pace, giving the player a time limit for choosing the columns in which to place them. A horizontal line stretching across the columns shows the maximum height of all of the piles of blocks, communicating to the player what should be minimized and how well they are doing, both visually and by their numeric score. This design is illustrated in Figure 1.

The game for solving scheduling problems consists of a succession of quick levels. The three different stages are played consecutively, with one tutorial level at the start of each stage. Jobs are randomly generated for each level, with a length of exponential distribution. Each level has a fixed number of jobs, and if a player arranges the jobs such that they all fit within the screen, the game allows the player to advance to the next level. If the player fails to appropriately arrange the jobs, she must restart a new game for that level. This generates new tasks of exponential length.

The tests were conducted in person. Players were asked to follow instructions provided by the video game without any verbal explanation. This is because we wanted to replicate the experience a player could obtain via a game obtained through digital distribution. After each play, the video game stored the resulting data in a private cloud.

### 3.3. Game 2: Facility Location Problem

In this section, we describe the implementation of the Facility Location Problem (FLP). First, we briefly provide FLP background and formulation. Second, we describe how we modeled FLP as a video game, outlining details of the game.

### 3.3.1. Background for Facility Location

Although the FLP is an NP-hard problem, it has many real-world applications such as refugee shelters [36], gas gathering station [37]. Nonetheless, for some instances of this problem, it is inherently intractable to find solutions in a reasonable time [38]. Using an optimal algorithm is unfeasible because of time constraints. Therefore, the FLP is approximated (with a significant error [39]) using heuristics or meta-heuristics. Thus, we will crowd-source solutions to the FLP.

FLP is about finding a set of facility locations so as to minimize the cost of satisfying a set of demand centers, while at the same time satisfying other constraints [40]. Each facility to be opened should be placed where a demand center is. Opening a facility comes with a cost, but so does serving a demand center distant from a facility or even operating the facility itself [41]. The goal is to minimize the total cost by finding the optimal placement of facilities according to the geographical distribution of a given set of demand centers [36,42].

Formally, FLP is defined in terms of the following symbols: $D$, the set of all demand centers; $F$, the set of all candidate facilities; $G$, the set of open facilities; $W$, the set of open facilities that did not fail; $f_i$, the cost of opening a facility at location $i$; $c_{ij}$, the cost of assigning demand center $j$ to facility $i$; $y_i$, a binary decision variable that indicates whether facility $i$ is open; $x_{ij}$, a binary decision variable that indicates whether location $j$ is assigned to facility $i$; $u_{ij}$, a binary decision variable that indicates whether location $j$ is assigned to facility $i$ after failures have occurred; and $v_i$, a binary decision variable that indicates whether open facility $i$ has failed.

We have modeled the FLP following Reference [43]. This method allows decision-makers to balance the number of facilities to open, the total distance before failure, and the total distance after failure.

1. The first problem consists of obtaining a total distance without failure:

    (a) $\min\limits_{y} \sum\limits_{i \in F} f_i\, y_i$

    (b) $\min\limits_{x} \sum\limits_{i \in F} \sum\limits_{j \in D} d_j\, c_{ij}\, x_{ij},$

where the first objective function minimizes the cost of opening facilities, and the second one minimizes the total distance without failures. These two objectives allow decision-makers to understand the impact of one objective over the other (i.e., how opening one more facility reduces the total distance).

The problem becomes robust (RFLP) when a solution is required to perform well, even upon failure of some installed facilities. The worst-case total distance after disruptions is multi-objective and binary; it is formulated as follows:

2. Worst-case total distance:

    (a) $\min\limits_{v} \sum\limits_{m \in G} f_m\, v_m$

    (b) $\max\limits_{u} \sum\limits_{k \in W} \sum\limits_{j \in D} d_j\, c_{kj}\, u_{kj}$

    subject to:

    i. $v_m \geq y_m, \ \forall m \in G$

    ii. $v_m \in \{0,1\}, \ \forall m \in G$

    iii. $u_{kj} \leq x_{kj}, \ \forall k \in W, j \in D$

    iv. $\sum\limits_{k \in W} u_{kj} = 1, \ \forall j \in D$

where the first objective function minimizes the number of facilities that fail. The second objective function maximizes the distance after failures. Constraint i ensures that only open facilities can fail. Constraints ii ensures that v is a binary decision variable. Constraint iii ensures that demand centers are re-assigned to open facilities, and constraint iv ensures that demand centers are re-assigned.

Finally, the trade-off between distance before failure and distance after failure is defined as follows.

3. Trade-off problem:

    (a) $\min\limits_{x} \sum\limits_{i \in F} \sum\limits_{j \in D} d_j\, c_{ij}\, x_{ij}$

    (b) $\min\limits_{u} \sum\limits_{k \in W} \sum\limits_{j \in D} d_j\, c_{kj}\, u_{kj}$

    subject to

    i. $\sum\limits_{k \in W} \sum\limits_{j \in D} d_j\, c_{kj}\, u_{kj} < \sum\limits_{n \in T} \sum\limits_{j \in D} d_j\, c_{nj}\, x_{nj}$

where the first objective function minimizes the distance before failure and the second objective function minimizes the distance after failure. Constraint i ensures that the distance before the failure of any additional trade-off solution is smaller than the distance after the failure of a solution found for the total-distance-without-failure problem. The objective of this problem formulation is to obtain more trade-off solutions for each number of facilities to open.

### 3.3.2. Datasets

We used two datasets: Swain and London. They respectively have 55 and 150 demand centers. Demand centers are represented by a set of coordinates in a Cartesian plane $(x, y)$ and a weight associated representing the center proportional demand. The higher the value of the weight, the higher the associated demand. Each demand center may hold a facility. To compute the overall distance, we consider the distance between each open facility and its corresponding demand centers. In our case, we considered the weighted Euclidean distance. The Swain data set gives the distribution of air travelers by destination for Washington, DC in 1960 [44], while the London dataset represents the location of gasoline stations and fire stations; there, distances are based upon a road network [45].

### 3.3.3. Game 2 Description, Facility Location

The FLP game is modeled after previous work on the robust facility location problem (RFLP) using genetic algorithms [43], where the RFLP is formulated as three different optimization problems. At a high level, these sub-problems are as follows:

- Minimizing both the cost of opening facilities and the cost of traveling between the facilities and demand centers; this is a multi-objective formulation of the FLP.
- Finding the worst possible failure combination, namely, the set of open facilities that result in the highest total distance should they fail, determining how each solution proposed in the previous step will perform as a solution to the RFLP.
- Finding optimal solutions to the RFLP, which involves a trade-off between the optimal solution when no facilities fail and its optimal value after facility failures. This sub-problem provides a human decision-maker with the ability to trade-off for a given domain.

The game is designed to address the FLP in two stages, played in consecutive order. On each stage, there is a category of levels, where each level corresponds to a single instance of the problem. The details are as follows:

- The player proposes solutions for instances of the FLP. A solution consists of a set of facilities that the player chooses to open.
- The player is presented with solutions provided by other players, and the player is asked to improve the total cost. Players must choose a set of open facilities; however, in this case, the game gives them a starting point.

In a match, opening a facility has an associated cost; nevertheless, to compute each candidate solution, we use a single score for total score of the players, incorporating both the total distance and opening facilities. This implicit assumption that the cost of opening a facility is equivalent to the cost of some distance may bias the players, making them more likely to focus their search with a fixed number of open facilities. The number of open facilities minimizes the total cost, leaving much of the solution space unexplored, and the game designer determines the cost for each open facility placed. Some levels include a limit on the number of facilities that the player can open, which is chosen at random. Limiting the number of facilities that a player can open encourages the player to explore every possible number of open facilities and to provide reasonable solutions along the entire Pareto boundary. Making this distinction among separate stages enables us to complement the genetic algorithm because the crowd-computing-based stage can complement those output solutions not found by the genetic algorithm.

#### User Interface Design

To keep the game independent of different domains in which instances of the problem may arise, we chose to employ abstract representations of the demand centers, facilities, weights, and assignments, which were free of any references to cities, vehicles, factories, or any other real-life element of these

instances, as shown in Figure 2. The visualization for the FLP consists of a 2D white canvas that displays demand centers as circular spots in the coordinates specified by the dataset. Each demand center may have a facility on it, displayed as a black square inscribed in the circle. All demand centers are shown in blue; however, their saturation varies with their weight—centers with low demand are a lighter blue while centers with a high demand are dark blue, as depicted in Figure 2.

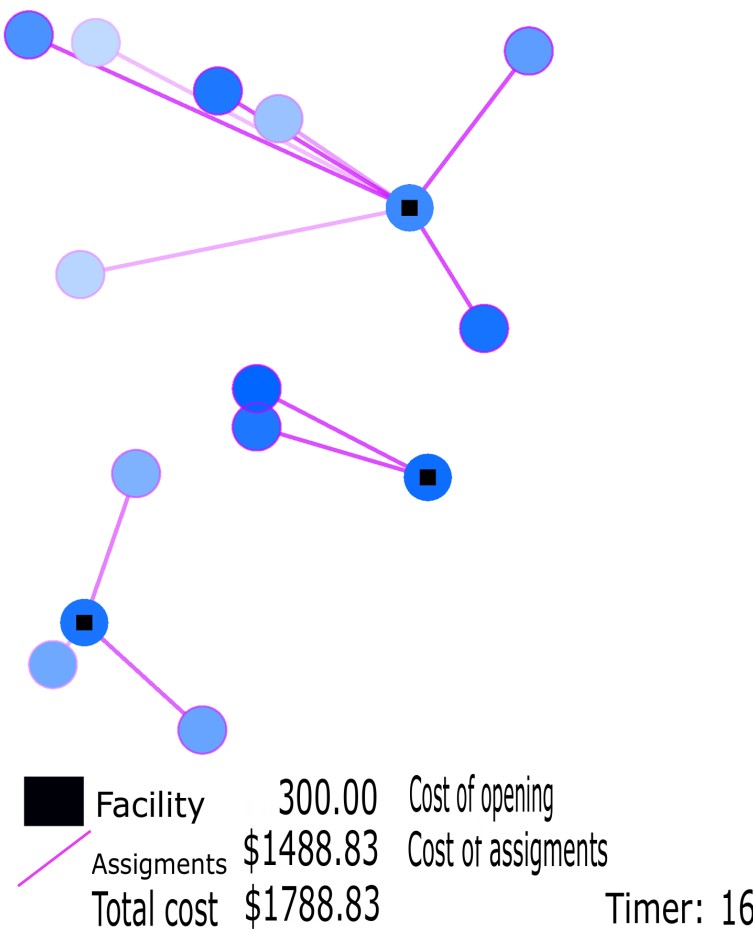

**Figure 2.** User interface for the Facility Location Problem (FLP) game. Circles represent demand centers and squares facilities (note facilities are placed in a demand center. All demand centers are colored in blue; color saturation varies with the demand center weight, representing the level of demand: the lighter the color, the lesser the level. Assignments between a demand center and a facility are represented by a straight line connecting them.

A straight line connecting two circles represents an assignment between a demand center and a facility. The line color saturation reflects the cost of the assignment, based on a function of the distance between the center and the facility, and the weight of the demand centers. Unassigned demand centers have blinking borders to attract the attention of the players.

For the game stages, the player needs to choose where to open a facility within a demand center. To open a facility, the player needs to tap an empty demand center. To close a facility, the player needs to tap an occupied demand center. The game is designed similar to a check-box control for standard graphical UIs. For the stages involving failed facilities, the player needs to manually assign a demand center to a facility. To achieve this, a player must tap-and-drag from the facility at the demand center,

forming a line connecting the two points; a line connects the facility to their moving finger (or cursor), showing how they are stretching the influence over a facility to cover new territory.

Gameplay and Playtesting

The game for solving the FLP consists of a quick succession of levels, progressing through the three different stages. There are two types of levels: tutorials and real data. A tutorial teaches the player the mechanics of the game, and for real datasets, players provide candidate solutions. A dataset contains a set of demand centers and their corresponding location and weight; the weight represents the magnitude of their demand, and the coordinates present the demand and location in the $(x, y)$ plane. However, in some stages, the players interact with candidate solutions provided by other players; thus, the levels can also include a set of opened facilities, a map assigning demand centers to a facility, or a set of facilities that are failing.

A tutorial level ends when the player completes the objective of the game being taught. Because it is not possible to determine if a player has found the best solution for an arbitrary dataset (which would require solving the optimization problem in the first place), the dataset levels end with a time limit. Finally, this approach grants the player the reward of advancing through the game and avoiding the frustrating experience of being stuck on a level that may be impossible to improve further.

As for the scheduler game, players were asked to follow written instructions provided by the games and were not given a verbal explanation, aiming to replicate the experience a player would obtain from a digital distribution on their device, unassisted. We observed the gameplay, and the server stored the resulting data. For these playtests, each of the games consisted of the level sequences described below.

Level Sequence for the FLP Game

First, the player goes through ten quick tutorial levels, each teaching one aspect of the FLP or a particular user interface interaction or game condition:

1. opening facilities
2. closing facilities
3. supplying demand centers
4. the cost of large distances
5. the weights of demand centers
6. the impact of weights in assigning a demand center to a facility
7. choosing the best spot for a single facility
8. having more than one facility open at the same time
9. the cost of having many facilities open
10. choosing the best spots to open multiple facilities, with a time limit

## 4. Results and Discussion

This section presents the results we have obtained throughout our experimentation. We present and discuss our results for the scheduling problems first, and for the Facility Location Problem next.

### 4.1. Scheduling Problems

For our first experiment, we have selected the following algorithms: LIST, Tabu Search, Greedy Best First and Genetic Algorithm, ordered from the less to the more powerful. The rationale behind algorithm selection is as follows: we picked the LIST algorithm for it is a well-known and straightforward heuristic that assigns jobs to the least busy machine [46]; the implementation of it can be found over the Internet, especially designed to tackle the Identical Parallel Machines (IPM) problem. We picked Tabu search, an algorithm that has been found to provide a problem solution in a reasonable time [47]; we developed a home implementation of Tabu search and applied it to both the IPM problem and the Job Shop Scheduling (JSS) problem. For dealing with JSS, we also used the

greedy best-first search (GBF) algorithm, which has been found to also provide a problem solution in a reasonable time. Finally, for the Open Shop Scheduling (OSS) problem, we used GBF and a popular genetic algorithm. See Figure 3, for a quick summary of this.

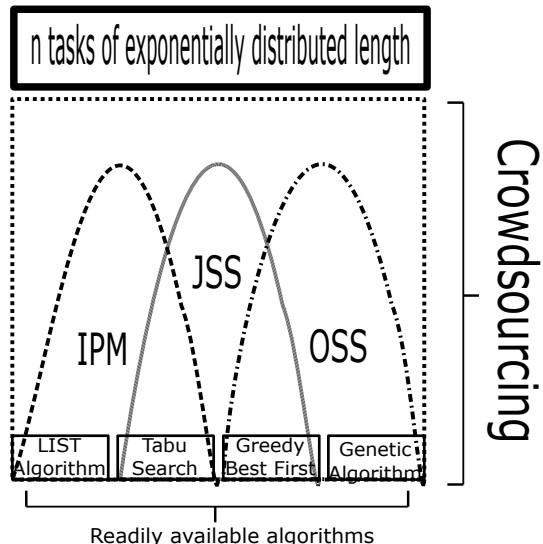

**Figure 3.** Experimental setup for evaluating all the variants of the scheduling problem, namely: Identical Parallel Machines (IPM), Job Shop Scheduling (JSS) and Open Shop Scheduling (OSS).

We have run all these algorithms 20 times, one for m = 5 and the other for m = 7, and picked the best result on each case. As shall be discussed later on in the text, each problem was solved via our crowdsourced video games by 20 different players, two times also, one for m = 5 and the other for m = 7. To validate our hypothesis (that users are up to solve computationally hard problems, complementing the solutions found by an algorithm) we have combined the output of a crowdsourcing (CS)-based method with that of various algorithms, yielding an overall method improvement as a result.

We have considered the makespan as an optimality criterion for all variants of the scheduling problems. For comparison purposes, we have selected the competitive ratio, this is because it is commonly used to express the performance of an online algorithm. The competitive ratio for an online algorithm is the quotient between the performance of it and the performance of an offline algorithm, which is determined analytically. However, because we cannot describe the behavior of a human player via a closed-form expression, we have approximated this ratio by complementing the best results obtained by players with the best results obtained by an algorithm.

Recall that to compare the performance by representing output results as whiskers boxes, we must consider the following characteristics:

- A small box represents a better performance.
- In our experiments we take the makespan as a performance measure (minimizing the total time to attend all tasks). Understanding as good performance:

    - The whisker box must be near to the lower limit.
    - The Whisker box must be compacted, that is, thin.

### 4.1.1. Solving IPM, Using Tabu, LA and Crowdsourcing (CS)

Figure 4 displays the results of solving IPM for m = 5 (see Figure 4a) and for m = 7 (see Figure 4b). Each whisker box represents the output solutions for the following approaches; first, we show Tabu search, second the crowdsourcing (CS) approach, third the LIST algorithm (LA), fourth the complement of CS-Tabu and finally a hybrid approach CS-LA.

Figure 4 illustrates a known weakness of the LA (which is mitigated by merging CS-LA), which shows up when tasks follow an exponential distribution. The weakness is as follows: each

task is always assigned to the least busy machine; since all machines tend to have a similar workload, then this results in a peak total time when a very long task is generated. In contrast, human players were always capable of planning and keeping a machine available in order to manage this eventuality, hence, the solutions of players had a much lower variance and fewer outlier cases, even when the median case was slightly higher.

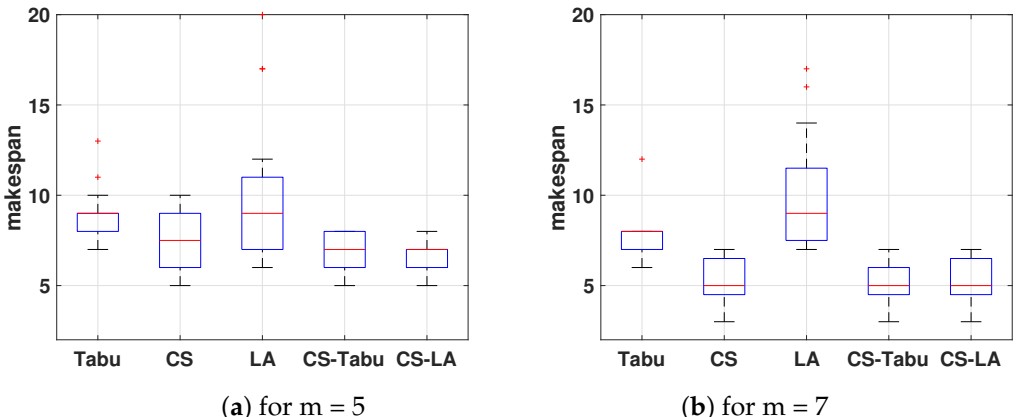

**Figure 4.** Performance of Tabu search algorithm, human players (crowdsourcing (CS)), LIST algorithm (LA), and the merged approaches (CS-Tabu and CS-LA) for solving the Identical Parallel Machines (IPM) scheduling problem for (**a**) m = 5, (**b**) m = 7 machines.

We have used a more sophisticated algorithm, such as the Tabu search algorithm, to validate our hypothesis as correct and also we want to show that by modeling the computationally hard problems as video games users can still yield competitive solutions, as these solutions can be used to complement those obtained mechanically. Figure 4 also shows the solution space for the Tabu search algorithm, notice it obtains smoother solutions than LA and crowdsourcing (CS), yet, the Tabu algorithm can still be complemented by some crowdsourcing solutions as visually shown in the figure (forth whiskers box, CS-Tabu). The Tabu search outperforms the LA, as it will always finds nearly optimal solutions, however, it is still outperformed after merging the crowdsourced output solutions with those obtained by it. From the forth whiskers box, we notice that the median goes into lower output values in both cases, namely, m = 5 and m = 7, besides the dispersion of the makespan values is smoother, meaning better approximations are found by combining these two methods.

From the fifth whiskers box (CS-LA) in Figure 4a,b (particularly in Figure 4a) we can observe that by merging the best solutions of the LA and those obtained with the crowdsourcing approach, corresponding to the new makespan value, the median is slightly lower than that obtained by the crowdsourcing-based approach, which clearly was the one that outperformed all approaches. Moreover, the range of output solution values decreases significantly, which leads to a smaller variance (variety of solutions) and zero outlier cases. Therefore, it is possible to significantly improve the makespan solutions by combining these two methods. Clearly having a hybrid approach outperforms individual methods, no matter if they are simple or complex. We have used seven results from the LA and thirteen results from crowdsourcing (CS) for Figure 4a (m = 5) and for Figure 4b (m = 7), we have used five solutions of the LA and fifteen solutions from crowdsourcing. However, using the crowdsourcing-based approach entails a cost, including the runtime and allocation of resources for computing the output solutions given by an algorithm and we shall also consider the time it takes users to complete a playthrough.

### 4.1.2. Solving JSS, Using Tabu, GBF and Crowdsourcing (CS)

Figure 5 displays the output results JSS after solving with diverse approaches for m = 5 (see Figure 5a) and for m = 7 (see Figure 5b). Each whisker box represents the output solutions for the

following approaches, namely, Tabu search, second the crowdsourcing approach, third greedy best first (GBF) algorithm, and finally the fourth and fifth whiskers boxes illustrate the hybrid approach CS-Tabu and CS-GBF.

Figure 5a shows that even though players have an stochastic behavior while solving complex scheduling problems some results are still considered as a complement to GBF and Tabu algorithms. For m = 7, the crowdsourcing-based on gamification in some cases exhibits a performance similar to that of GBF search algorithm, and could complement the worst answers given by the Tabu algorithm. It is because players think differently, they can adopt diverse strategies, resulting in excellent gameplay in some cases and poor gameplay in other cases. However, the correct solutions yielded by humans using the crowdsourcing approach are helpful when complementing the makespan values for the GBF approach for a larger problem instances.

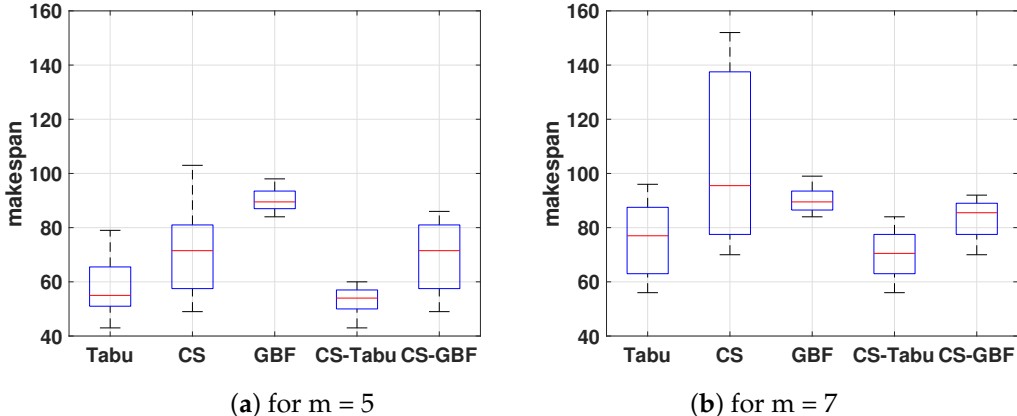

(**a**) for m = 5                  (**b**) for m = 7

**Figure 5.** Performance of Tabu search algorithm, crowdsourcing (CS), greedy best-first (GBF) search algorithm, and the merged approaches (CS-Tabu and CS-GBF) for solving the Job Shop Scheduling (JSS) for (**a**) m = 5 and (**b**) m = 7, respectively.

We have used the Tabu search algorithm in order to show that users can still complement it as shown in Figure 5a,b, in the forth whiskers box. These figures show improvement in the output makespan values as the median is slightly better and the overall whiskers box is closer to the inferior limit, meaning users can compete against powerful mechanically found approaches. Nonetheless, let us consider the following: a straight forward task would be that of inserting some solutions from the crowdsourcing approach into the GBF. Whilst, in contrast for inserting some solutions from the crowdsourcing approach into the Tabu algorithm—which is outside of the scope of this work—we would need considering the following problems, namely: first, how to extract each move a player made up until the solution he/she provided, second should it be given as initial solution or as an arbitrary iteration to the Tabu algorithm. Therefore, it is not straight forward task.

Likewise as done in Figure 4a,b, in order to obtain the fifth whisker box shown in Figure 5a,b, we have merged the best output solutions from GBF and crowdsourcing approaches, respectively. We have used four output solutions from the crowdsourcing approach and 16 solutions from the GBF search algorithm for Figure 5a, and 14 solutions from crowdsourcing and six solutions from GBF for Figure 5b. Both graphics show that the range of output solutions is smaller than those obtained alone by crowdsourcing. Moreover, although the range of CS-GBF solutions is greater than that of the GBF, the CS-GBF solutions are less than the median of the GBF, indicating a significant improvement. Additionally, approximately 75% of the makespan solutions are less than 81 and 89 for m = 5 and m = 7, respectively; in contrast, individual computed output solution values for the crowdsourcing and GBF approaches are slightly higher.

### 4.1.3. Solving OSS, Using Genetic Algorithm (GA), GBF and Crowdsourcing (CS)

Figure 6 displays the output results JSS after solving with diverse approaches for m = 5 (see Figure 6a and for m = 7 (see Figure 6b. Each whisker box represents the output solutions for the following approaches, namely, genetic algorithm (GA), second the crowdsourcing approach, third the GBF algorithm, and finally the fourth and fifth whiskers boxes illustrate the hybrid approach CS-GA and CS-GBF.

The first whiskers box from Figure 6a,b, shows the output results obtained using a traditional genetic algorithm. Clearly the genetic algorithm approach outperforms all other approaches, for bigger instances, m = 7 it cannot be complemented with those output solutions obtained using crowdsourcing, as Figure 6b illustrates. Nevertheless, users have yielded as good enough solutions as the genetic algorithm, for some cases, as Figure 6a illustrates. Yet, inserting crowdsourcing-based solutions to a genetic algorithm is not a straight forward task; consider the following assumptions, namely: first, we must have knowledge of exactly which moves the player made; second within the genetic algorithm we need to find the exact intermediary state where to insert the solution; third, clearly identify non-dominated solutions so it is worth injecting them into the genetic algorithm, which already is an NP-hard problem.

On the other hand, Figure 6b (for m = 7), we can argue that due to the robustness of the genetic algorithm, as we expected the resulting output solutions of the genetic algorithm yielded better results than those yielded by the crowdsourcing approach. However, genetic algorithm does many iterations to reach an objective function, while crowdsourcing participants have only played once in order to obtain an equivalent solution, justifying why genetic algorithm obtains better solutions, as we fixed the number of iterations to 100. To improve our crowdsourcing-based approach and the output solutions we would have to train the players; implying each participant plays a greater number of games and spends a greater overall time to learn optimization techniques.

In contrast when complementing a not so robust approach, as Figure 6a,b illustrates the crowdsourcing (CS)-based approach has obtained better scores than some of the GBF algorithm. The (GBF) has difficulties to find solutions because this algorithm is less likely to sacrifice an immediate gain for a more significant long-term gain, in contrast to the behavior of the human players (even though stochastic) who identified these situations and have followed the path that led them to a better overall score (Figure 6a), without having to explore all possible moves.

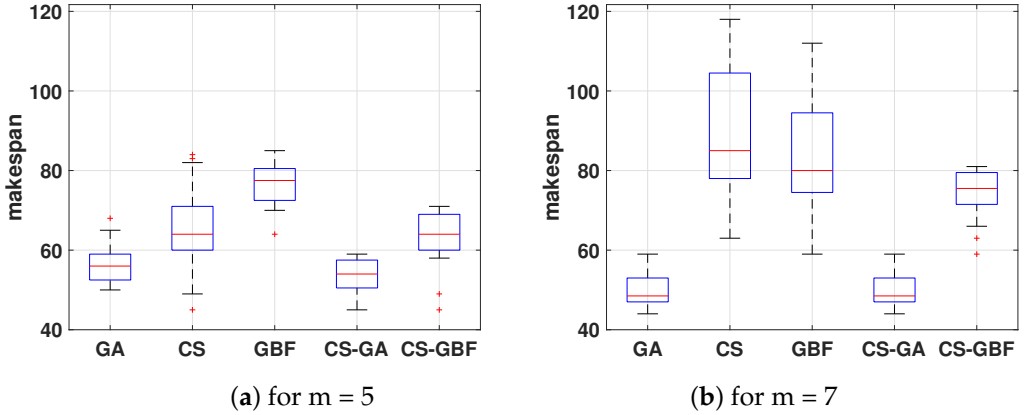

(**a**) for m = 5          (**b**) for m = 7

**Figure 6.** Performance of genetic algorithms (GA), crowdsourcing (CS), greedy best-first (GBF) search algorithm, and the merged approaches (CS-GA and CS-GBF) for solving the Open Shop Scheduling (OSS) for (**a**) m = 5, (**b**) m = 7 machines.

Similar to the previous figures, we have added a fifth whisker plot for GBF-CS in Figure 6a,b, we have obtained these results by selecting the best output performance values from GBF search algorithm and from crowdsourcing-based approach. For instance, for m = 5 in Figure 6a, we have considered four solutions from GBF and 16 solutions from crowdsourcing. The range of solutions is smaller than that for crowdsourcing (CS) but greater than that for the GBF. This finding does not imply that the GBF solutions are better than the other solution sets, as more than 75% of the GBF-CS solutions have a lower makespan than the GBF solutions (second box) and 100% of the GBF-CS solutions have a lower makespan than 75% of the crowdsourcing (CS) solutions (second whisker box). In addition, we built the fifth whisker plot in Figure 6b for m = 7, using 11 GBF and nine crowdsourcing values respectively. We can observe that the variability of the solutions is significantly lower than that for the crowdsourcing and GBF solutions. This finding suggests that the overall solutions have similar results, in contrast to the individual crowdsourcing and GBF solutions, where the solution variability is substantial.

We can conclude that the output solutions reported here obtained via gamification techniques after modeling different variants of the scheduling problem have yielded good approximations, as these problems are still hard for computers to solve efficiently. Why can we reach that conclusion? Because the experimental results show that users can complement from simple algorithms such as LIST and GBF, to more sophisticated algorithms such as Tabu search and finally even robust algorithms such as a genetic algorithm. Note that, for this study, video game players have only played the different instances of scheduling problems (IPM, JSS, and OSS) twice; after players second try we have recorded their results. We have demonstrated, in this pilot study, that even if users play for a few times, they yield relatively reasonable approximate solutions to simple variants (IPM), and to more complex variants of scheduling problems (OSS and JSS). For example, consider output solutions obtained for the OSS by means of a genetic algorithm and by means of crowdsourcing-based approach, if we compare the operation of the genetic algorithms, which require large numbers of iterations to give an acceptable solution, we can observe that the crowdsourcing approximates to those solutions, yet, with a much smaller number of attempts. In addition, we are not yet considering that users improve as they increase the number of times they play because we want to model our gamification techniques similar to genetic algorithms.

## 4.2. FLP

We have performed four experiments, two of them focus on solving the FLP using the Swain dataset and the other two use the London dataset. For the first two experiments, we solve using genetic algorithms (NSGA-II and MOPSDA) as authors report in Reference [43], and then we use the crowdsourcing-based approach. We have carried out experiment three and four with a larger number of facilities (London dataset) and also we solve using genetic algorithms and the crowdsourcing approach as we have resumed in Figure 7. All four experiments were carried out for 20 times, that is, we have executed 20 different runs of the genetic algorithms, and we have asked 20 different players to solve each problem once (one time for the Swain dataset and another for the London dataset) to obtain a significant statistical sample.

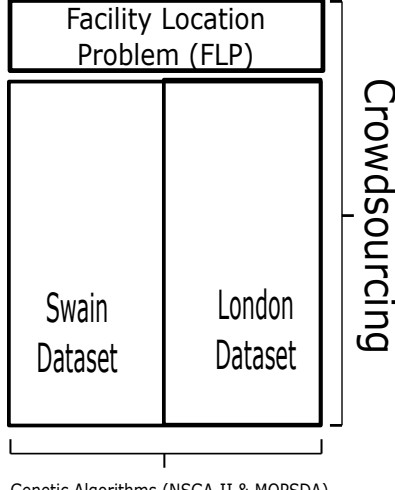

**Figure 7.** Experimental setup for evaluating the Facility Location Problem, using two datasets (Swain and London) commonly found in literature benchmarks.

### 4.2.1. Results for the Swain Dataset

The FLP solution requires the identification of the Pareto set, a set of solutions that cannot improve one objective without harming another objective. We evaluate the aggregate player data by measuring how well the data provides an approximation to the Pareto set, as obtained by other methods.

Figure 8a,b, illustrate the output results we have obtained using genetic algorithms and CS-based approach, respectively. We have found that the users yield better results in many cases, as shown in Figure 8c, for example, finding the best location to place facilities for $n = 1, 2, 3, 4, 5, 6, 7, 8, 9, 10, 11, 12, 13, 14, 15$. Sometimes, the crowdsourcing-based approach finds solutions whereas genetic algorithms do not, for example, when $n = 13$, the distance found by players is *distance* = 1898.49, while the genetic algorithm gives no value. genetic algorithms do not find an approximate solution to the Pareto front when it gets stuck at a local minimum, genetic algorithms cannot differentiate between a local minimum and a global minimum; as there is no mathematical proof for a genetic algorithm to converge to the global optimum. Figure 8a shows the best solutions genetic algorithm obtains after executing 20 different runs reason why it seems that some solutions are dominated by others, yet the reason is that we obtained these solutions separately.

Figure 8b also shows that the output solutions yielded by players provide better approximation solutions when allocating a large number of facilities. For example, for $n = 42, 49, 50, 51, 52, 53$, and 54. Therefore, similar to the case of having a small number of facilities, users provide complementing solutions to genetic algorithms. In addition, the CS-based approach explores a more thorough solution space, finding solutions for $n = 38, 39, 40, 41$, whereas genetic algorithms show gaps or give no solutions. Genetic algorithms do not have a mean to prove that the optimal solution has been found, genetic algorithms only give good approximations for NP-hard problems.

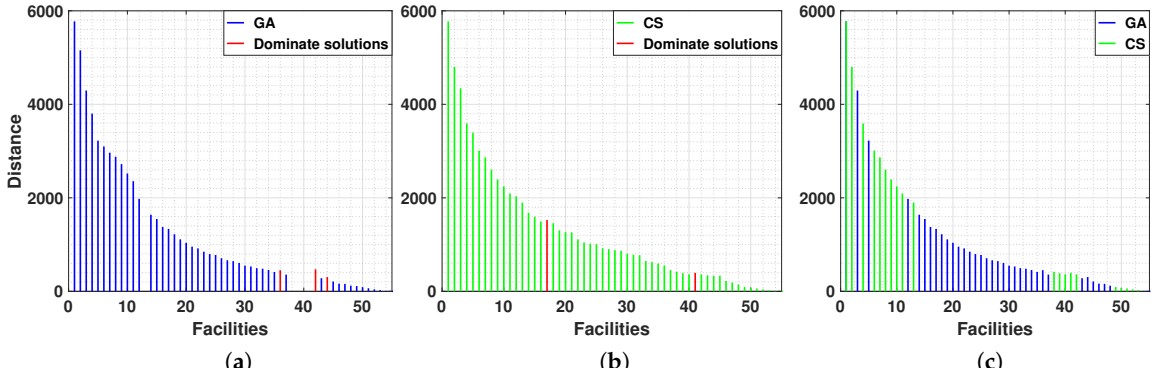

**Figure 8.** The Pareto set after solving the facility location problem for the Swain dataset using: genetic algorithms (**a**), crowdsourcing (**b**) and finally the genetic algorithm complemented by crowdsourcing solutions (**c**).

We have performed a third experiment, using an approximation to obtain missing facilities for the FLP. With such aim, and after 20 different executions of the genetic algorithm, we have used the data gathered along with cubic splines for data interpolation. This method is useful for finding missing sets of data; for this purpose, we have used a polynomial for approximation. Even when comparing the crowdsourcing-based approach to this method, we found that the crowdsourcing output solution to the Pareto set space yields for the missing facilities a generally superior (or better approximation) than those cubic splines yielded, as shown in Figure 9.

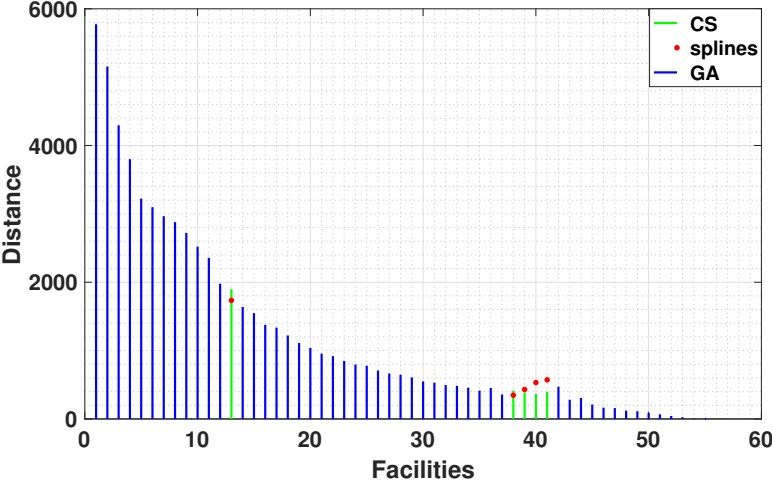

**Figure 9.** The Pareto set after solving the facility location problem for the Swain dataset using: genetic algorithms, crowdsourcing and cubic splines.

### 4.2.2. Results for the London Dataset

We performed two other experiments for solving the FLP using a larger dataset (London dataset). Figure 10a shows the Pareto set distribution found using genetic algorithms. Likewise, Figure 10b shows the Pareto set distribution found by crowdsourcing (CS) the problem to users. Again, the problem was solved for two conflicting objective functions. We have selected the best answers computed from 20 different players (shown in Figure 10b), and 20 different executions of the genetic algorithms (shown in Figure 10a). Figure 10c, illustrates how the solutions of players complement those found using genetic algorithms.

From the data we obtained shown in Figure 10a,b we found that users obtained better solutions in many cases, as shown in Figure 10c, for example, in finding the best locations to place facilities for $n = 5$ up until $n = 27$. Sometimes, the CS approaches find a solution, whereas genetic algorithms do

not, for example, when $n = 2, 3, 4$ or the genetic algorithms give no value, as for $n = 28$ to $n = 42$. Finally, the genetic algorithms present gaps from $n = 98$ to $n = 119$.

Note that the total distance should always decrease as the number of facilities increases (because the best solution for $n$ open facilities, when augmented with any other open facility, will yield a lower total distance for the $n + 1$). However, in the crowd-sourced data, the graph presents regions in which the total distance increases because those solutions were independently encountered and were found separately in the raw data. Through some necessary processing, statistical analysis resulted in an upper bound to the total distance of the solution located to the right for any given case.

Figure 10 also shows that the players give better solutions when placing a large number of facilities or when asked to identify the best places for facilities. For instance, for $n = 120$ to $n = 155$; the crowdsourcing (CS) approach finds a better solution than the genetic algorithm.

Figure 11 shows the genetic algorithm solutions for the London dataset case, where we show how genetic algorithms can be complemented by two different methods, namely, crowdsourcing and interpolation based on cubic splines. In this figure, analogous to Figure 9, we illustrate that in most cases, when complementing the missing solutions for the genetic algorithms, the crowdsourcing solutions are better than those obtained by interpolation. We chose one of the best cubic splines tools for interpolating by polynomials; however, once again, the crowdsourcing solutions are the best approximations.

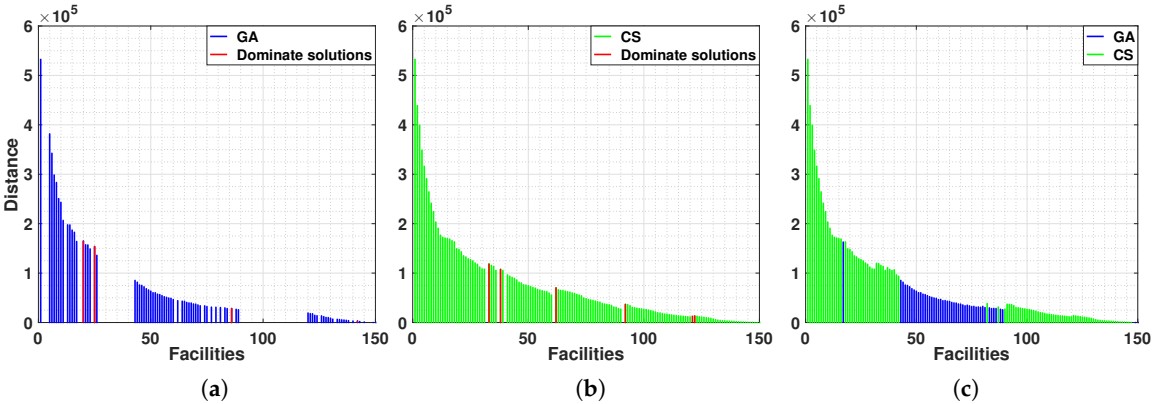

**Figure 10.** The Pareto set after solving the facility location problem for the London dataset using: genetic algorithms (**a**), crowdsourcing (**b**) and the genetic algorithm complemented by crowdsourcing solutions (**c**).

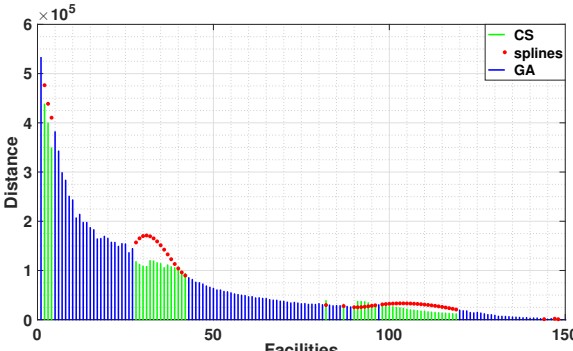

**Figure 11.** The Pareto set after solving the facility location problem for the London dataset using: genetic algorithms, crowdsourcing and cubic splines.

From Figures 8 and 10 we can conclude that users can complement the Pareto search space; in fact, in many cases, the solutions obtained by the crowdsourcing approach are better than those obtained with the genetic algorithms. It would be interesting to explore a hybrid combination of

these approaches because genetic algorithms present gaps, and our empirical study has shown that crowdsourcing can provide a sufficient number of solutions for filling those gaps. Likewise, with the help of machine learning techniques, it may be possible to analyze the heuristics that users follow or implement to solve computationally hard problems. We propose that combining some of the best solutions of the players with genetic algorithms can decrease the convergence time of genetic algorithms or increase the quality of the solutions; however, decreasing the convergence time of genetic algorithms is beyond the scope of this work.

Our primary finding is that users can obtain better solutions (in some instances) in the Pareto set than previous genetic algorithm-based methods, and thus, the gaps in a GA-based approach can be filled the best crowd-sourced solutions. Initial experiments and pilot tests with the FLP modeled as a video game demonstrate that the solution quality obtained by genetic algorithms can be improved, even for experiments with large population sizes (large number of facilities). A detailed analysis of the heuristics of users remains as future work because we must first investigate and accurately quantify various aspects of this hybrid approach.

Humans can master optimization because their insight enables them to find better places for facility allocations in a visual mode. In addition, users can find solutions in seconds, whereas computers may can take hours. We found that the players solved the FLP levels in 40 s, whereas a genetic algorithm requires 2–3 h to solve the same problem.

### 4.3. Complementing Genetic Algorithms with Crowdsourcing-Based Solutions

Both the crowdsourcing approach based on video games and genetic algorithms aim to find the Pareto set for a given multi-objective optimization problem. Difficulty arises because genetic algorithms present gaps in the solution space, whereas the crowdsourcing approach performs a more thorough search of the solution space; however, the results show that the genetic algorithms obtain better results for some areas of the Pareto set. To address this issue, we propose measuring the hypervolume as a quality indicator and validating the hypothesis that the solutions obtained by crowdsourcing are of high quality.

The hypervolume indicates the area that covers an approximation to the Pareto border given by an algorithm. A reference point delimits the area for each objective function. A larger hypervolume indicates a better convergence and better coverage for the optimal Pareto border.

Figure 12a, shows the hypervolume computed for the Swain dataset, based on 20 playthroughs of different users and 20 different executions of the genetic algorithm based on NSGA-II with MOPSDA [43]. We computed the hypervolume for the solutions from the crowdsourcing approach and for the solutions from the genetic algorithms. Figure 12a shows that genetic algorithms have a larger hypervolume value than the one computed for the crowdsourcing approach because larger hypervolume values mean to convergence towards the true Pareto front of a problem, and these specific genetic algorithms compute near optimal solutions and get stuck in a local minimum for dominated solutions.

Figure 12b shows the computed hypervolume for the London dataset, based on 20 playthroughs performed by different users and 20 executions of the genetic algorithm based on NSGA-II with MOPSDA [43]. This figure illustrates a substantial improvement for the crowdsourcing approach; in this case, the approach finds solutions that are similar or superior to those found with the genetic algorithm. This finding validates the hypothesis that the insight and intuition of users can be leveraged for solving MOPs; in addition, the responses of players complement the solution space obtained by genetic algorithm, thus producing a complete Pareto set.

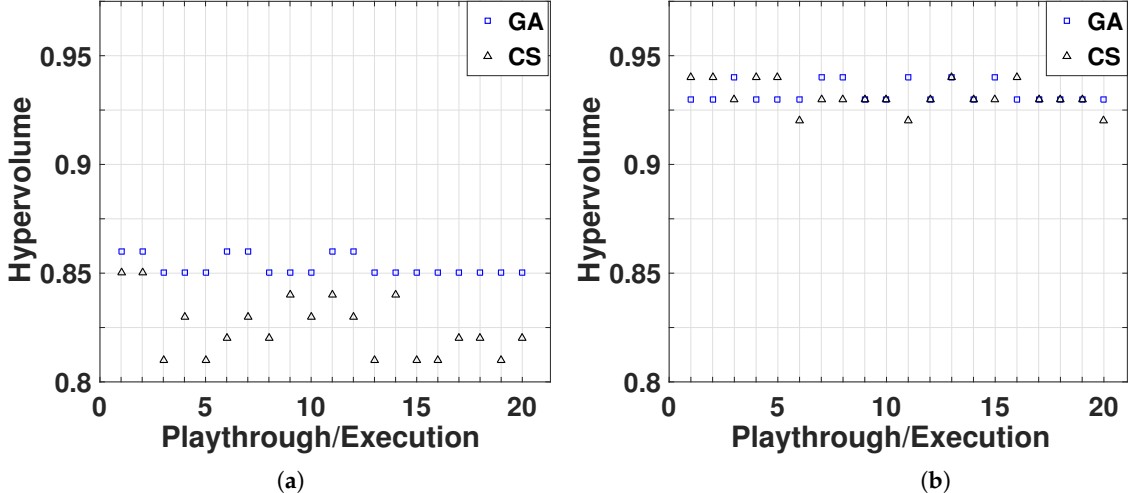

**Figure 12.** Computed hypervolume after solving the Facility Location Problem (FLP) by using the crowdsourcing approach and genetic algorithms (GA) for Swain (**a**) and London (**b**) datasets.

In addition to the previous conclusions, the crowdsourcing approach is valuable for several reasons:

- Decision-makers can rapidly obtain solutions and identify the best trade-off between the number of open facilities and total distance.
- Optimal answers obtained from the crowdsourcing approach can complement previous solutions or those obtained by genetic algorithms; as future work, we will consider a hybrid approach.
- Players learn multi-objective optimization by merely playing a video game.
- Initially, the results, particularly the computed hypervolume values for the Swain dataset, did not seem promising. However, further results obtained for the London dataset indicate that the crowdsourcing approach is scalable; and thus, users can solve NP-hard problems for significant applications.
- A possible milestone for solving NP-hard problems is given (in a fast manner compared with traditional methods based only on heuristics, meta-heuristics, and genetic algorithms). However, more work is needed for verification, as traditional methods may show a reduced convergence time.

Although the hypervolume shows improvement in solution quality as modest (though significant enough for large instances of the problem), the advantage of the crowdsourcing technique over genetic algorithm methods is the straightforwardness applying it (arising from the absence of any need to train or knowledge of multi-objective optimization from users). Crowdsourcing computationally hard problems for users to solve can significantly improve other techniques in terms of computational complexity, and convergence time.

We have obtained 0.87 as hypervolume value for the complement shown in Figure 8c, for the Swain dataset, which clearly outperforms both genetic algorithm and crowdsourcing individual solution values. We have obtained 0.95 for the hypervolume value after computing the complement solution we obtain in Figure 10c, for the London dataset, again we the complement approach outperforms both genetic algorithm and crowdsourcing individual solution values. Therefore, we obtained a better solution.

### 4.4. Analysis of Results for Swain and London Datasets

We ran the Friedman test to the results we have obtained after running different tests for genetic algorithms (GA), crowdsourcing (CS) and the complement approach (GA-CS). The Friedman test compares all methods (GA, CS and GA-CS) and tells if there exists statistical significant differences between the mean ranks of the methods compared.



Friedman's test tells whether there are general differences, but it does not indicate which particular methods differ from each other. There was a statistical significant difference in the perceived distance depending on the type of method used whilst solving the FLP. As results we have for the Swain dataset $X_r^2 = 25.75, p = 0.00001$ and for the London dataset $X_r^2 = 24.01, p = 0.00001$. Hence, we can see that there is an overall statistically significant difference between the mean ranks while comparing the three methods.

In order to be certain if our combined GA-CS method presents a statistical significant difference while computing the Pareto set and to complement findings from the hypervolume, we have applied the Wilcoxon signed-rank test for different combinations of the methods. Post hoc analysis with Wilcoxon signed rank tests was performed with a Bonferroni correction applied, which resulted in a significance level set at $p < 0.017$ and 95% confidence level. We note there was a statistically significant reduction in the perceived distance in the genetic algorithm versus the genetic algorithm supplemented with crowdsourcing solutions (GA-CS) for the Swain dataset ($Z = -4.8599, p = 0.00001$) and ($Z = -2.8405, p = 0.00452$) for the London dataset, respectively.

## 5. Conclusions and Future Work

In this paper, we have proposed to solve some NP-hard problems such as the Scheduling Problems in diverse variants (Identical Parallel machines, Job Shop Scheduling and Open Shop Scheduling) and the Facility Location Problem (FLP). Besides after giving evidence that humans leverage their insights and intuition for solving complex problems, we provide a new method based on combining human outputs with does of diverse algorithms from readily available to robust solutions. After supplementing some algorithms we view an overall makespan value reduction of 75% (for simple optimization problems, with single objective). Confirming true our hypothesis, hence, addressing multi-objective optimization problems. For the FLP we have observed that there exists a significant statistical difference for the methods we evaluated, whilst genetic algorithms provide an approximation to the Pareto front, we can observe that the combined method provided a better hypervolume value for both Swain and London datasets 0.87 and 0.95, respectively. To confirm such evidence we have drawn statistical tests (Friedman and Wilcoxson with Bonferroni adjustment), which show a significant statistical difference between the methods measured as reported above.

### 5.1. Lessons Learned from Game Design

We have learned, from this study, that there is a need for interactive demos for the user to easily adapt to the game and know how to improve their score. Most of the players, regardless of their age and gender, entertained themselves by playing our video games; even though the only requirement they had was to try to get the best possible score. It is a great challenge to design a video game that is not going to be used for entertainment purposes only, but also for the development of skills. Although not many people who played our video game had an academic background in mathematics or computing, they managed to learn and adequately solve complex problems.

What would we do different? One of the possible disadvantages of using video games to model computationally hard problems is that sometimes users do not read the tutorials; players get stuck and get frustrated. In many games, pop-ups appear in which video game designers, being as clear as possible, teach some game functionality to the user. We believe it is best to use pop-ups instead of presenting the user a tutorial per level. Also, to improve engagement of players, we could give a reward to those players that obtained the best scores.

For users to get engaged playing video games, sometimes it is not enough to challenge them to beat the score other players obtain; to motivate players to play more we could give away different game backgrounds as a reward for obtaining the best scores. We could also change the game controls to something more modern, for instance, instead of having buttons to swap tasks between machines; we could just allow users to use their fingers and drag the tasks between machines and place the task on the machine selected by the users.

We could also incorporate another kind of video game; a multi-player game. For the robust facility location problem, a player can be the defender, incorporating a defense mechanism into the facilities making them indestructible; another player may be the attacker, also having a mechanism but in this case of attack, where he could destroy facilities that are not protected by the defender. Therefore, this game allows decision-makers to trade-off when there are failures in the facilities and to take the corresponding actions.

*5.2. Lessons Learned from Crowdsourcing*

One of the first things is to confirm that the size of the crowd is irrelevant as long as the goal is achieved. In our case, success comes because we have successfully confirmed our hypothesis, at such aim we have shown that the combined method outperforms individual human based and mechanically based methods.

*5.3. Limitations for the Crowdsourcing Approach*

A limitation for the crowdsourcing-based approach is that not all answers provided by users are significant, not all players understand at glance the video game. Since we carried out this research in a university facility, having diverse public (not all related to engineering or mathematical backgrounds) a lack of expertise in multi-objective optimization problems (MOPs) can lead to low quality solutions. Even with such limitations we found statistical significant improvement regardless that not all players contribute with reasonable solutions.

Another drawback but difficult to measure (subjective) is the time to model and develop the video games. For an experienced person (programming and with knowledge to MOPs) can take a couple of weeks to code to an inexperienced person can take months. Nonetheless, once the game has been modeled and developed it is easy to reuse for other datasets of interest and it can be used as a simulation model that require fast response for emergency situations.

*5.4. Future Work*

For future work we are going to classify human strategies using machine learning techniques, also we will be carrying out data mining to figure out what do winner playthroughs have in common and also to see if players follow the same steps (do these steps resemble an algorithm? If so which one?).

Moreover, future research will involve extracting strategies from players. This requires to have a proper formalization of a match, so as to identify the strategy, if any, a player follows to get a high-score. Then, we also have to take into account player cheating, as in Reference [48], where authors have identified ways players cheat such as using online tutorials, using code only meant for developers, and even hacking, in order to obtain high scores. Additionally, we must consider ways of comparing standard strategies for solving, say facility location problem, with those we could have obtained via gamification and machine learning. This will definitely shed light on our conclusion, as found by Reference [49], who used linear regression to express differences between problem solving styles and problem solving approaches.

**Author Contributions:** Conceptualization, M.V.-S., J.E.R.-M. and C.Z.; Formal analysis, M.V.-S., J.E.R.-M. and R.M.; Investigation, M.V.-S., D.A.L.-V. and H.Z.; Methodology, M.V.-S. and Huaxing Zhu; Software, M.V.-S. and D.A.L.-V.; Supervision, J.E.R.-M., R.M., C.Z. and D.A.L.-V.; Validation, J.E.R.-M., R.M., C.Z. and D.A.L.-V.; Writing—original draft, M.V.-S., H.Z., and D.A.L.-V.; Writing—review & editing, J.E.R.-M., C.Z. and R.M. All authors have read and agreed to the published version of the manuscript.

**Funding:** No funding was received for this project.

**Acknowledgments:** The authors want to thank specially to all those players that collaborated in this investigation, to all of the undergraduate and graduate students of Universidad Autónoma Metropolitana, Unidad Cuajimalpa.

**Conflicts of Interest:** The authors declare no conflict of interest. The funders had no role in the design of the study; in the collection, analyses, or interpretation of data; in the writing of the manuscript, or in the decision to publish the results.

## Abbreviations

The following abbreviations are used in this manuscript:

| | |
|---|---|
| CS | Crowdsourcing |
| FLP | Facility Location Problem |
| GA | Genetic Algorithm |
| GBF | Greedy Best First |
| IPM | Identical Parallel Machines |
| JSS | Jop Shop Sheduling |
| LA | LIST Algorithm |
| MOEA | Multi-Objective Evolutionary Algorithm |
| MOPs | Multi-objective Optimization Problems |
| OSS | Open Shop Scheduling |
| RFLP | Robust Facility Location Problem |
| UI | User Interface |

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
