# Peer review of "Complementing Solutions to Optimization Problems via Crowdsourcing on Video Game Plays"

_applsci, doi:10.3390/app10238410_

Round 1

Reviewer 1 Report

The paper seems to be nice but I could not really realize whether the novelty of the paper is enough or not since it is not so near my topic. However, English should be improved; as an example, Line 39--40 of Page 2 is not clear (replace "can help solve" with "can help to solve").

Author Response

Dear reviewer, 

Please find attached the responses to your comments. 

Best regards

Reviewer 2 Report

The process of optimization of problems and video games is very relevant. Although research related to optimization is frequent, considering video games to improve the process arises not frequently. In fact, the authors do not include any bibliographical references on the subject.

Going in deep on the topic video games and problem-solving would be useful. For example, among many others

Hamlen, K. R. (2017, 2018/07/01). General Problem-Solving Styles and Problem-Solving Approaches in Video Games. Journal of Educational Computing Research, 56(4), 467-484. https://doi.org/10.1177/0735633117729221 

Hamlen, K. R., & Blumberg, F. C. (2015). Chapter 4 - Problem Solving Through “Cheating” in Video Games. In G. P. Green & J. C. Kaufman (Eds.), Video Games and Creativity (pp. 83-97). Academic Press. https://doi.org/https://doi.org/10.1016/B978-0-12-801462-2.00004-7 

The paper is difficult to read, especially a would recommend do not use so many abbreviations. Moreover, in general terms, the research design is not clear and must be improved. Even, what kind of cognitive process it is supposed that must be involved in the paly process of the game. Maybe figures could help to make it more clear.

Considering my background, social sciences, it is difficult for me to give you recommendations in the data analysis.  

Author Response

Dear reviewer, 

Please find attached the replies to your comments. 

Best regards. 

Round 2

Reviewer 2 Report

Thanks for your review. I value your effort following the previous recommendations. Objectives have been reformulated, and small changes have been introduced. The paper is more precise than the earlier version. I hope the suggestions have been helpful. I include the file considering the differences between the two versions. Even you know them, maybe it could be useful.
